# Animal Models for the Study of Neurological Diseases and Their Link to Sleep

**DOI:** 10.3390/biomedicines13082005

**Published:** 2025-08-18

**Authors:** Carmen Rubio, Emiliano González-Sánchez, Ángel Lee, Alexis Ponce-Juárez, Norma Serrano-García, Moisés Rubio-Osornio

**Affiliations:** 1Neurophysiology Department, National Institute of Neurology and Neurosurgery “Manuel Velasco Suárez”, Mexico City 14269, Mexico; mrubio@innn.edu.mx (C.R.); nasmy_sr@yahoo.com.mx (N.S.-G.); 2Medicine School, Autonomous University of Chiapas, Tapachula Campus, Tapachula 30700, Mexico; emiliano.gonzalez04@unach.mx; 3National Institute of Public Health, Cuernavaca 62100, Mexico; 4Neurochemistry Department, National Institute of Neurology and Neurosurgery “Manuel Velasco Suárez”, Mexico City 14269, Mexico; alexis.packe2001@gmail.com

**Keywords:** sleep fragmentation, neurodegenerative models, oxidative stress, neuroinflammation, mitochondrial dysfunction

## Abstract

Sleep is a vital biological function governed by neuronal networks in the brainstem, hypothalamus, and thalamus. Disruptions in these circuits contribute to the sleep disturbances observed in neurodegenerative disorders, including Parkinson’s disease, epilepsy, Huntington’s disease, and Alzheimer’s disease. Oxidative stress, mitochondrial dysfunction, neuroinflammation, and abnormal protein accumulation adversely affect sleep architecture in these conditions. The interaction among these pathological processes is believed to modify sleep-regulating circuits, consequently worsening clinical symptoms. This review examines the cellular and molecular mechanisms that impair sleep regulation in experimental models of these four disorders, emphasizing how oxidative stress, neuroinflammation and synaptic dysfunction contribute to sleep fragmentation and alterations in rapid eye movement (REM) sleep and slow-wave sleep (SWS) phases. In Parkinson’s disease models (6-OHDA and MPTP), dopaminergic degeneration and damage to sleep-regulating nuclei result in daytime somnolence and disrupted sleep patterns. Epilepsy models (kainate, pentylenetetrazole, and kindling) provoke hyperexcitability and oxidative damage, compromising both REM and SWS. Huntington’s disease models (R6/2 and 3-NP) demonstrate reduced sleep duration, circadian irregularities, and oxidative damage in the hypothalamus and suprachiasmatic nucleus. In Alzheimer’s disease (AD) models (APP/PS1, 3xTg-AD, and Tg2576), early sleep problems include diminished SWS and REM sleep, increased awakenings, and circadian rhythm disruption. These changes correlate with β-amyloid and tau deposition, glial activation, chronic inflammation, and mitochondrial damage in the hypothalamus, hippocampus, and prefrontal cortex. Sleep disturbances across these neurodegenerative disease models share common underlying mechanisms like oxidative stress, neuroinflammation, and mitochondrial dysfunction. Understanding these pathways may reveal therapeutic targets to improve both motor symptoms and sleep quality in neurodegenerative disorders.

## 1. Introduction

Sleep is regulated by a complex network of brain regions, including the brainstem, hypothalamus, thalamus, and cerebral cortex, which interact through specific neural circuits to control the sleep–wake cycle [1,2]. GABAergic and galaninergic neurons in the ventrolateral preoptic nucleus (VLPO) of the hypothalamus suppress wake-promoting areas such as the locus coeruleus and tuberomammillary nucleus, facilitating sleep onset and maintenance [3,4]. Multiple neurotransmitter systems, including gamma-aminobutyric acid (GABA), glutamate, acetylcholine, orexin, noradrenaline, norepinephrine, serotonine and histamine, modulate sleep architecture through precisely balanced excitatory and inhibitory synaptic pathways [5,6,7]. The degeneration of hypocretin-producing neurons in the lateral hypothalamus contributes to narcolepsy and the sleep disturbances commonly observed in neurodegenerative disorders such as Parkinson’s and Alzheimer’s disease [8,9]. In Parkinson’s disease, alterations in the mesocorticolimbic dopaminergic system disrupt circadian rhythms and sleep architecture, leading to insomnia, periodic limb movements, and excessive daytime somnolence [10,11]. Epilepsy involves the dysfunction of the thalamocortical networks that normally regulate sleep spindles, potentially promoting epileptiform discharges during non-REM sleep [12,13,14]. Huntington’s disease (HD) features circadian rhythm disturbances and the altered expression of clock genes such as Per1 and Bmal1, associated with neurodegeneration in the suprachiasmatic nucleus and other sleep-regulating regions [15,16,17]. In Alzheimer’s disease, β-amyloid and tau aggregation in the hypothalamus, brainstem, and prefrontal cortex correlates with sleep fragmentation and reduced SWS and REM sleep activity, thereby accelerating disease progression [18,19]. These findings highlight that sleep regulation involves specific neuronal and molecular mechanisms that become disrupted in major neurological disorders. Oxidative stress, neuroinflammation, mitochondrial dysfunction, and abnormal protein accumulation impair sleep-regulating circuits in neurodegenerative disease models. Sleep is a fundamental biological process essential to brain homeostasis, neuronal repair, and cognitive function, making it a critical subject of study in the context of neurological health. In neurodegenerative diseases, sleep disturbances are not merely secondary symptoms but active contributors to disease progression. Despite the high prevalence of sleep disorders in conditions such as Parkinson’s disease, epilepsy, Huntington’s disease, and Alzheimer’s disease, there is a lack of comprehensive studies examining how sleep-regulating circuits are affected in neurological experimental models. Present research mainly focuses on clinical symptoms or behavioral observations, with limited exploration of the cellular and molecular mechanisms underlying sleep disruption in animal models. This review addresses that critical gap by analyzing how oxidative stress, neuroinflammation, mitochondrial dysfunction, and synaptic impairments contribute to sleep fragmentation and alterations in REM and slow-wave sleep phases. Unraveling these interconnected mechanisms is crucial for developing targeted therapies that not only address motor and cognitive deficits and restore healthy sleep architecture. Improving sleep quality may represent a powerful strategy to mitigate neurological decline and enhance the overall quality of life of patients with neurodegenerative disorders, especially considering how these pathways disrupt sleep regulation in experimental models of Parkinson’s disease, epilepsy, Huntington’s disease, and Alzheimer’s disease.

We conducted a comprehensive literature search using Google Scholar and PubMed to identify relevant studies examining sleep’s role in experimental models of neurodegenerative diseases, specifically Parkinson’s disease, Alzheimer’s disease, epilepsy, and Huntington’s disease. The search employed multiple keyword combinations including SWS and REM, glymphatic system, protein clearance, β-amyloid, tau, α-synuclein, mutant huntingtin, oxidative stress, neuroinflammation, synaptic plasticity, signaling pathways, Nrf2, NF-κB, GABA, and glutamate. We included experimental studies (in vivo and in vitro) and review papers that utilized established models of the specified neurodegenerative disorders. Studies were selected based on their methodological quality, scientific rigor, and relevance to the physiological and molecular mechanisms through which sleep facilitates neuroprotection and disease modulation. We analyzed the selected literature to identify both the common and disease-specific mechanisms by which sleep influences protein aggregation, oxidative and inflammatory responses, neurotransmitter regulation, and the activation of survival and plasticity-related signaling pathways. We synthesized these findings to better understand sleep’s role in mitigating neurodegenerative processes.

## 2. Sleep and the Diseased Brain: Experimental Models in Neurodegenerative and Neurological Disorders

### 2.1. Models of Parkinson’s Disease and Sleep Disruptions

In Parkinson’s disease, experimental models such as 6-hydroxydopamine (6-OHDA) and 1-Methyl-4-phenyl-pyridine (MPP+) have proven essential for investigating dopaminergic neurodegeneration and associated sleep disturbances. Neurodegeneration directly affects the neuronal networks responsible for sleep–wake control, including the locus coeruleus, tuberomammillary nucleus, lateral hypothalamus, and suprachiasmatic nucleus. This leads to sleep fragmentation, reduced SWS, and the altered REM sleep patterns commonly observed in Parkinson’s patients [20,21,22,23,24]. Mitochondrial dysfunction and oxidative stress, central features of Parkinson’s disease that are replicated in these models, impair sleep by damaging critical neurons and altering the electrical excitability necessary for normal sleep architecture [25,26,27,28,29]. MPP^+^, the toxic metabolite derived from MPTP, blocks mitochondrial complex I, increasing reactive oxygen species (ROS) generation, which causes oxidative damage to neuronal proteins, lipids, and DNA in brain regions controlling sleep regulation [30,31,32,33,34,35]. Similarly, 6-OHDA induces severe oxidative stress through autooxidation in dopaminergic neurons, promoting apoptosis and glial activation, which intensify neuroinflammation and disrupt circadian rhythms [36,37,38,39,40]. Abnormal α-synuclein aggregation, a hallmark of Parkinson’s disease, correlates with sleep disturbances. These aggregates appear early in the brainstem nuclei involved in sleep regulation, as evidenced by the strong association between REM sleep behavior disorders and disease progression [41,42,43,44]. Oxidative stress and impaired cellular degradation systems promote protein aggregation [45,46,47,48], creating a vicious cycle that affects both neuropathology and sleep-related clinical symptoms. Disruptions in calcium homeostasis, excitotoxicity, and axonal transport, worsened by mitochondrial dysfunction and oxidative stress, damage orexinergic neurons in the lateral hypothalamus [7]. These neurons are crucial for maintaining stable wakefulness, and their impairment contributes to excessive daytime sleepiness and sleep fragmentation in both animal models and patients [22,49,50,51]. Neuroinflammation triggered by activated microglia and astrocytes leads to pro-inflammatory cytokine release (IL-1β, TNF-α, IL-6), which disrupts sleep-associated neurotransmitter regulation, worsening sleep fragmentation and reducing REM sleep [52,53,54,55,56]. Numerous drugs, including dopaminergic medicines like levodopa and dopamine agonists (like pramipexole and ropinirole), melatonin, and clonazepam, have been researched in this model to treat sleep–wake disorders, especially REM sleep behavior disorder [57,58]. These therapies try to change the circadian and REM-related circuits that are impacted in Parkinson’s disease or restore dopaminergic tone.

Genetic mutations in SNCA (α-synuclein), *PARK2*, *PINK1*, *DJ-1*, and *LRRK2* affect protein aggregation and mitochondrial function, contributing to Lewy body formation and non-motor symptoms, including sleep disturbances [59,60,61,62]. Additionally, polymorphisms affecting oxidative stress and inflammatory responses may influence sleep disruption severity, motor symptoms and disease progression [63,64]. Dysfunction in signaling pathways including MAPK, NF-κB, and Nrf2, which are essential for oxidative stress responses and neuroinflammation, affects neuronal survival in sleep-regulating nuclei and influences sleep–wake cycle disturbances in Parkinson’s disease [65,66,67,68,69].

The 6-OHDA and MPTP models effectively recapitulate both the characteristic dopaminergic neurodegeneration of Parkinson’s disease and its non-motor symptoms, particularly sleep disturbances. These models demonstrate how oxidative stress, mitochondrial dysfunction, calcium dysregulation, neuroinflammation, and protein aggregation impact sleep-regulating neural circuits. They provide valuable experimental platforms for investigating pathogenic mechanisms and evaluating therapies targeting both motor and non-motor symptoms [70,71,72,73] (Figure 1).

### 2.2. Epilepsy Models and Sleep Architecture

Animal models have been essential for understanding epilepsy pathophysiology, epileptogenesis, and therapeutic development. Kainate (KA) and pentylenetetrazol (PTZ) seizure models effectively replicate various epileptic seizures, enabling a detailed investigation of the underlying cellular and molecular mechanisms and associated sleep disorders. These models serve as vital tools for understanding the complex relationships among the key of components neuronal dysfunction, oxidative stress, and sleep architecture in human epilepsy, particularly in chronic and treatment-resistant cases.

The kainate model employs an AMPA/KA receptor agonist that induces severe seizures and neuronal damage, primarily in the hippocampus, mimicking human temporal lobe epilepsy [74,75,76]. Glutamatergic excitotoxicity causes intracellular calcium overload, activating enzymes like phospholipase A2 and nitric oxide synthase. This leads to excessive reactive oxygen and nitrogen species production, causing significant oxidative damage and cell death through necrosis and apoptosis [77,78,79,80].

This model demonstrates how metabolic alterations lead to functional sleep changes, including reduced SWS, diminished REM sleep, and increased sleep fragmentation. These changes result from neuronal damage in critical regions such as the hippocampus, thalamus, and cortex, accompanied by inflammatory and oxidative processes [81,82,83,84]. The KA model enables causal investigation of epilepsy, oxidative stress, and sleep disruption from both molecular and functional perspectives.

The PTZ model induces generalized seizures by antagonizing GABA_A_ receptors, disrupting excitatory–inhibitory balance and triggering neuronal hyperexcitability [85,86]. This hyperactive state promotes ROS formation through oxidative enzyme activation and mitochondrial dysfunction, compromising cellular integrity [87,88]. The resulting oxidative stress response affects antioxidant protein production (superoxide dismutase and catalase) and activates the NF-κB-controlled inflammatory pathways involved in epileptogenesis [89,90,91]. The PTZ model shows sleep architecture changes characterized by reduced sleep efficiency, decreased REM sleep, and prolonged sleep latency. These modifications relate to alterations in key neurotransmitters (serotonin and dopamine) and increased oxidative stress [92,93,94]. By improving GABAergic tone and controlling circadian rhythms, drugs like benzodiazepines (such as diazepam and clonazepam) and melatonin have been investigated as ways to improve the quality of sleep in these models [95,96]. Although their effects on sleep can vary depending on the dosage and treatment length, antiepileptic medications such as levetiracetam and valproate have also demonstrated the ability to alter sleep architecture while reducing the frequency of seizures [97].

The kindling paradigm uses repetitive subconvulsive electrical stimulation in limbic areas like the hippocampus or amygdala, providing a crucial tool for investigating progressive epileptogenesis and associated sleep architecture changes [98,99,100,101]. This model demonstrates substantial decreases in SWS and REM sleep, with notable increases in sleep fragmentation [102,103,104], disrupting sleep quality and restorative capacity. These sleep disorders correlate with kindling-induced neuronal hyperexcitability, driven by neurotransmitter dysregulation including increased glutamate release and GABAergic system dysfunction [105,106]. At the molecular level, repetitive stimulation activates signaling pathways such as MAPK and JNK, promoting synaptic plasticity and hyperexcitability. Kindling also markedly increases ROS production, causing oxidative damage to lipids and DNA while compromising mitochondrial function, exacerbating epileptogenesis [107,108,109,110]. Oxidative stress disrupts sleep architecture by triggering inflammatory processes through microglial activation and pro-inflammatory cytokine production, affecting circadian rhythms [111,112]. These processes alter the neurotransmitters governing the sleep–wake cycle, leading to fragmentation and reduced SWS and REM sleep phases. Sleep deprivation impairs memory consolidation and neuronal repair, intensifying hyperexcitability and promoting chronic epilepsy progression [113].

Optogenetic models have revolutionized epilepsy research by enabling precise neuronal population control through light stimulation in genetically modified animals [114,115]. Prolonged excitatory neuron activation or inhibitory neuron suppression can trigger seizures, promote abnormal plasticity, and alter functional connectivity, directly affecting sleep architecture and quality [116]. The optogenetic stimulation of specific circuits, including the lateral hypothalamus and ventrolateral preoptic nucleus, substantially alters sleep duration and quality, reducing REM sleep and promoting prolonged wakefulness [117,118,119]. These sleep abnormalities create a detrimental cycle where sleep disturbance intensifies neuronal hyperexcitability and seizure frequency. These models reveal cellular-level calcium homeostasis disruption, crucial in neuronal excitotoxicity during seizures. This alteration increases ROS generation, causing oxidative damage that compromises neuronal function [114]. Oxidative stress exacerbates neuronal inflammation and synaptic dysfunction, also affecting circadian rhythms and sleep architecture [113].

Research on genetic epileptic channelopathy models has revealed significant sleep architecture and regulation changes. Animal models with ion channel gene mutations (*SCN1A* for sodium, *KCNQ2* for potassium, or *CACNA1H* for calcium) show marked reductions in SWS, increased sleep fragmentation, and prolonged nocturnal wakefulness, suggesting impaired thalamocortical circuits and sleep-regulating structures, including the hypothalamus [120,121,122,123]. These changes demonstrate the strong relationship between abnormal neuronal excitability and sleep–wake cycles, indicating that channelopathic mutations not only cause seizures but also disrupt sleep homeostasis from early developmental stages [124]. These models help understand how chronic ionic dysregulation affects sleep physiology through both direct and indirect pathways, including oxidative stress, mitochondrial dysfunction, and neuroimmune activation processes [125,126,127,128]. These processes may worsen sleep problems by creating pro-inflammatory environments in the brain regions essential for circadian and sleep–wake control (Figure 2).

### 2.3. Huntington’s Disease Models and Sleep Disruptions

Sleep disturbances represent early and persistent manifestations of Huntington’s disease (HD), often appearing before motor symptoms and highlighting their potential as clinical biomarkers and therapeutic targets [15,129]. These abnormalities include reduced sleep quality, decreased REM and SWS, increased sleep fragmentation, prolonged sleep latency, and circadian rhythm disruption [130].

HD animal models, including transgenic mice such as YAC128 and R6/2, as well as knock-in animals, demonstrate substantial alterations in sleep architecture and circadian rhythms. These changes encompass sleep fragmentation, reduced SWS, decreased overall sleep duration, and disrupted circadian gene expression in critical brain regions including the suprachiasmatic nucleus, lateral hypothalamus, and striatum [15,131,132,133]. R6/2 mice expressing mutant huntingtin fragments exhibit progressive sleep–wake cycle disturbances, characterized by significant reductions in REM and SWS, increased sleep fragmentation, and prolonged sleep onset times [134]. These sleep alterations correlate with dysfunction in the hypothalamic and brainstem nuclei governing circadian rhythm and sleep regulation, where oxidative stress and neuroinflammation cause neuronal impairment [131,134,135,136,137]. Several pharmaceutical treatments have been tested to lessen sleep disturbances in these HD models. While sedative antidepressants like trazodone have been investigated for the treatment of insomnia [138], melatonin and its receptor agonists have demonstrated promise in enhancing sleep quality and re-establishing circadian rhythms [139]. Compounds that target adenosinergic transmission or anti-inflammatory drugs like minocycline are also being studied for their dual function of lowering neuroinflammation and regulating sleep [140]. Animal models, including transgenic sheep and monkeys, have been developed and show circadian dysfunction with more complex phenotypes, enhancing their translational relevance for evaluating sleep restoration therapies [141,142,143,144]. However, these models are costly and require extended periods for symptom manifestation.

Toxin-induced models using 3-nitropropionic acid (3-NP), an irreversible mitochondrial succinate dehydrogenase inhibitor, show substantial sleep architecture disruptions like R6/2 mice, including reduced REM sleep, increased awakenings, and sleep fragmentation [145,146]. These disturbances are linked to glial inflammation, mitochondrial dysfunction, and oxidative stress in sleep-regulating brain regions such as the hypothalamus and brainstem [147,148]. Quinolinic acid administration, a neurotoxic tryptophan pathway metabolite causing excitotoxicity, disrupts sleep cycles by altering neurotransmitters and promoting neuroglial inflammation, leading to sleep fragmentation and circadian rhythm disturbances [149,150,151,152].

HD results from abnormal CAG repeat expansion in the *HTT* gene, producing a mutant huntingtin protein variant [153,154,155]. This abnormal protein triggers multiple pathogenic processes including mitochondrial dysfunction, oxidative stress, DNA damage, calcium homeostasis disruption, synaptic dysfunction, and apoptotic pathway activation [156,157]. Oxidative stress serves as a central element in HD pathology, characterized by elevated ROS, a reduced cellular antioxidant capacity, and oxidative damage to lipids, proteins, and nucleic acids [158,159,160]. These metabolic alterations affect brain regions controlling sleep, exacerbating circadian rhythm and sleep homeostasis problems. At the molecular and cellular levels, these models demonstrate that oxidative stress in sleep-regulatory regions impairs neurochemical signaling relevant to sleep–wake regulation, affecting critical neurotransmitters including GABA, glutamate, adenosine, and monoamines [137,161,162,163]. Mitochondrial failure generates excessive ROS, disrupting neuronal and glial homeostasis and promoting neuronal death in circadian rhythm-regulating areas [164,165].

Chronic microglial and astrocyte activation leads to pro-inflammatory cytokine release (IL-1β and TNF-α), disrupting synaptic transmission and sleep homeostasis [166,167]. This pro-inflammatory and oxidative environment alters circadian clock gene expression and function (*Clock*, *Bmal1*, *Per*, and *Cry*), directly affecting sleep–wake cycle synchronization [131,137,168]. Disrupted purinergic transmission, particularly reduced adenosine levels, a sleep-promoting compound, further exacerbates the fragmentation and poor sleep quality observed in these models [137,169,170]. In cellular models using neuronal and stem cell cultures, mutant huntingtin expression induces mitochondrial dysfunction, oxidative stress, impaired axonal transport, and toxic protein aggregate accumulation [171,172,173]. Mitochondrial damage, oxidative stress, excitotoxicity, glial inflammation, and circadian gene dysfunction collectively form a pathogenic network underlying HD sleep disorders. These findings emphasize the importance of targeting oxidative stress and neuroinflammation as potential approaches to improve sleep disturbances in this condition [164,166,174] (Figure 3).

### 2.4. Sleep as a Critical Factor in Alzheimer’s Disease

Evidence has highlighted the crucial role of sleep disruptions in Alzheimer’s disease etiology [175,176]. These disruptions appear as early clinical signs and significantly influence disease progression. Experimental models have been essential for understanding the cellular and molecular mechanisms linking sleep with key AD pathology, including amyloid-β accumulation, tau hyperphosphorylation, neuroinflammation, oxidative stress, and synaptic dysfunction [177,178,179].

Transgenic mouse models, including APP/PS1 and 3xTg-AD, have proven highly valuable for replicating the neuropathological features and sleep-related abnormalities observed in human patients [180,181]. In the 3xTg-AD model, chronic sleep deprivation for eight weeks caused significant memory deficits, increased tau phosphorylation, reduced synaptic protein levels (including PSD-95), and altered CREB expression, an essential transcription factor for synaptic plasticity [181,182]. Additionally, sleep deprivation triggers substantial astrogliosis, evidenced by elevated GFAP levels, suggesting that disrupted sleep intensifies glial activation and neuroinflammatory responses [183,184,185]. The 5xFAD model, characterized by rapid amyloid-β (Aβ) plaque accumulation, also demonstrates spontaneous sleep fragmentation. These mice show significant decreases in SWS and shorter, more fragmented sleep episodes, particularly in females, reflecting the sex-specific sleep disturbances observed in women with AD [186,187]. When 3xTg-AD mice undergo experimental sleep fragmentation, they display elevated hippocampal Aβ40 and Aβ42 levels, alongside intensified neuroinflammatory responses, indicating the increased vulnerability of the AD brain to sleep disturbances [188]. From a metabolic perspective, sleep fragmentation in AD models reduces mitochondrial activity. In 3xTg-AD mice, six weeks of sleep disruption downregulates mitochondrial biogenesis and electron transport chain complex expression, accompanied by decreased AMPK/SIRT1/PGC-1α pathway activity [189]. These changes result in a diminished antioxidant capacity and increased oxidative damage. However, the partial reversal of these alterations occurs with sleep recovery, highlighting the therapeutic potential of sleep normalization for restoring mitochondrial homeostasis [190]. Sleep disruption significantly affects tau pathology in experimental animals. PS19 mice expressing P301S mutant tau exhibit substantial sleep disorders, characterized by reduced SWS and REM sleep, a loss of REM atonia, and disrupted sleep spindles [191]. These changes closely resemble those seen in individuals with REM sleep behavior disorder, a prodromal feature of tauopathies. Pharmacological intervention using dual orexin receptor antagonists (e.g., DORA-12) restores normal sleep architecture in these animals, suggesting potential therapeutic approaches for early-stage intervention [18,192].

Melatonin supplementation, which has been demonstrated to increase SWS and lower oxidative stress, and GABAergic drugs like zolpidem, which promote sleep continuity, are further pharmaceutical strategies able to optimize sleep and lessen neurodegeneration in AD animals [193,194]. Orexin receptor antagonists and anti-inflammatory drugs are also being researched to address sleep issues and the neuroinflammatory pathways that contribute to the development of AD [195,196].

Animal studies demonstrate that even one night of sleep deprivation increases tau levels in the interstitial fluid and cerebrospinal fluid (CSF). This increase depends on neuronal activity and orexin signaling, as orexin inhibition reduces tau accumulation, supporting the concept that sleep deprivation promotes tau pathology spread [191]. Acute experimental models, including the intracerebroventricular (ICV) injection of Aβ oligomers or streptozotocin (STZ-ICV) administration, induce notable sleep disturbances. Aβ injection causes sleep fragmentation, microglial activation, and increased oxidative stress. The STZ-ICV model, which mimics sporadic AD and cerebral insulin resistance, shows cognitive impairments, circadian rhythm disturbances, and fragmented sleep, all associated with mitochondrial dysfunction and reactive oxygen species accumulation [197,198,199,200].

Non-mammalian models have provided valuable insights. In *Drosophila melanogaster*, neuronal Aβ expression alters circadian and sleep–wake cycles [201]. Transgenic AD models in *Danio rerio* (zebrafish) enable real-time sleep behavior observation in vivo, utilizing larval transparency and genetic manipulability [202]. These models offer unique perspectives on evolutionarily conserved sleep-regulating mechanisms and their relationship with neurodegenerative diseases. At the molecular level, experimental models show that sleep deprivation impairs the glymphatic clearance of metabolites, including Aβ, during SWS [203]. Disrupted sleep alters essential circadian clock gene expression (*BMAL1*, *CLOCK*, *PER2*), impairing the sleep–wake cycle and desynchronizing critical brain regions such as the suprachiasmatic nucleus [204]. Furthermore, chronic glial activation and inflammation from sleep deprivation create a neurotoxic environment that accelerates dementia progression [34,205]. These experimental models are crucial for both replicating AD neuropathology and validating the pathogenic and therapeutic significance of sleep disorders. They enable mechanistic studies of disease progression, the evaluation of pharmacological and genetic interventions, and the identification of early molecular biomarkers. These models demonstrate that maintaining sleep quality could serve as a powerful neuroprotective strategy against AD progression and provide an essential experimental framework for developing sleep-targeted therapeutics in neurodegenerative disorders (Figure 4).

## 3. Conclusions

The clearance of waste from the brain is significantly more efficient during sleep than during wake periods, as the convective clearance of neurotoxic compounds that accumulate during wakefulness is highly increased. This might boost brain restoration in many neurological disorders (Parkinson’s disease, epilepsy, Alzheimer’s disease, and Huntington’s syndrome). Their pathogenic mechanisms are all heavily influenced by sleep and include protein aggregation, mitochondrial dysfunction, oxidative stress, excitotoxicity, and neuroinflammation, despite their clinical heterogeneity. The glymphatic system is a primary pathway for CNS waste clearance during SWS, removing potentially harmful misfolded proteins (e.g., tau, α-synuclein, β-amyloid, and mutant huntingtin). The accumulation of harmful proteins plays a central role in causing neuronal damage in neurodegenerative diseases [189,203]. Additionally, sleep affects the expression of genes linked to energy consumption, neurogenesis, synaptic plasticity, and DNA repair, which all are crucial functions to maintain the integrity of neural networks and brain homeostasis [206,207]. Sleep has a profound influence on major cellular signaling pathways such as PI3K/Akt/mTOR, BDNF/TrkB, ERK/MAPK, and CREB, which are essential for memory consolidation, synaptic remodeling, and neuronal survival functions, which are impaired in these illnesses [208]. The progression and stabilization of neurodegenerative diseases are deeply influenced by the balance between NF-κB (a pro-inflammatory mediator) and Nrf2 (a key antioxidant regulator), especially during sleep. This balance is crucial: tipping it toward sustained inflammation and oxidative stress (as with chronic sleep loss) accelerates neurodegenerative processes [209], while restoring or maintaining it (via healthy sleep) offers neuroprotection and can help stabilize the progression of neurodegenerative diseases. Sleep helps lower seizure risk by enhancing GABAergic inhibition, which suppresses excessive neuronal firing and reduces glutamatergic excitation, lowering the chance that neurons will synchronize and “ignite” a seizure [12,210,211]. Many sleep-dependent processes are common denominators across brain diseases; improving sleep could have broad, multi-level therapeutic effects, reducing many damaging disease mechanisms at once. Focusing on sleep/wake regulation might be promising because sleep is a central “control hub” that influences many of the molecular and cellular pathways disrupted in neurological disorders. Sleep is a natural process in which the brain repairs, recalibrates, and clears debris. By targeting and improving specific aspects of sleep, some therapies might slow, stabilize, or even reverse the biological processes driving those diseases and represent a common therapeutic axis. Ongoing research into how sleep architecture is affected in each disorder might create unified, effective neurorestorative treatments that benefit many disorders and a shared opportunity for intervention across brain diseases (Table 1).

## Figures and Tables

**Figure 1 biomedicines-13-02005-f001:**
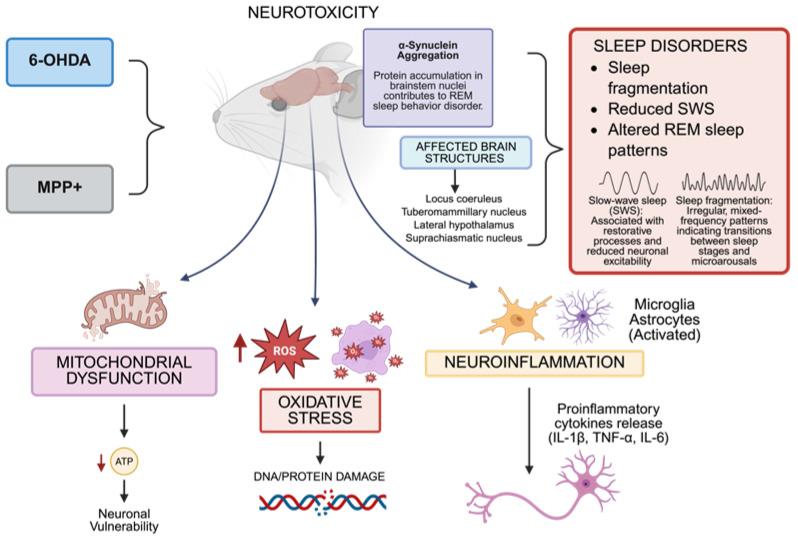
In murine models, the administration of neurotoxins, including 6-hydroxydopamine (6-OHDA) and 1MPP+, leads to neurotoxicity characterized by mitochondrial injury, reactive oxygen species (ROS), oxidative stress, and neuroinflammation. These changes impact important brain centers that control sleep, including the locus coeruleus, tuberomammillary nucleus, lateral hypothalamus, and suprachiasmatic nucleus, causing poor sleep quality, a decrease in slow-wave sleep (SWS), and changes in rapid eye movement (REM) sleep. The accumulation of misfolded α-synuclein in brainstem nuclei and the production of pro-inflammatory cytokines (e.g., IL-1β, TNF-α, IL-6) are the causes of dysfunction in sleep–wake-regulating circuits. The dates show ↓ = Decreased; ↑ = Increased (Created with https://BioRender.com). By Emiliano Gonzalez and Alexis Ponce, 2025.

**Figure 2 biomedicines-13-02005-f002:**
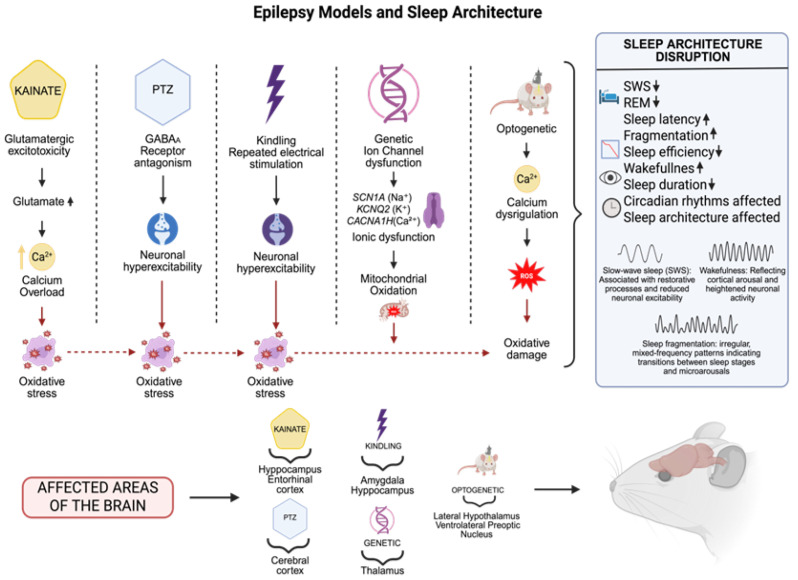
Experimental models of epilepsy replicate distinct seizure mechanisms: in the case of kainate, a glutamatergic agonist, (increases glutamate and serum calcium levels),temporal lobe seizures are induced at a cellular level by excitotoxicity; pentylenetetrazole (PTZ), a gamma-aminobutyric acid A (GABA_A_) receptor antagonist: generalized seizures; kindling: progressive epileptogenesis; genetic models: inherited epilepsies such as ion channel mutations; and optogenetics: manipulation of sleep circuits. These models affect important brain areas involving the hippocampus, entorhinal and cerebral cortex, amygdala, thalamus, and hypothalamus, and are accompanied by changes in sleep features (decreased SWS REM, sleep efficiency, sleep duration, enhanced fragmentation, prolonged latency, altered circadian rhythm and sleep architecture) The dates show ↓ = Decreased; ↑ = Increased (Created with BioRender.com). By Emiliano Gonzalez and Alexis Ponce, 2025.

**Figure 3 biomedicines-13-02005-f003:**
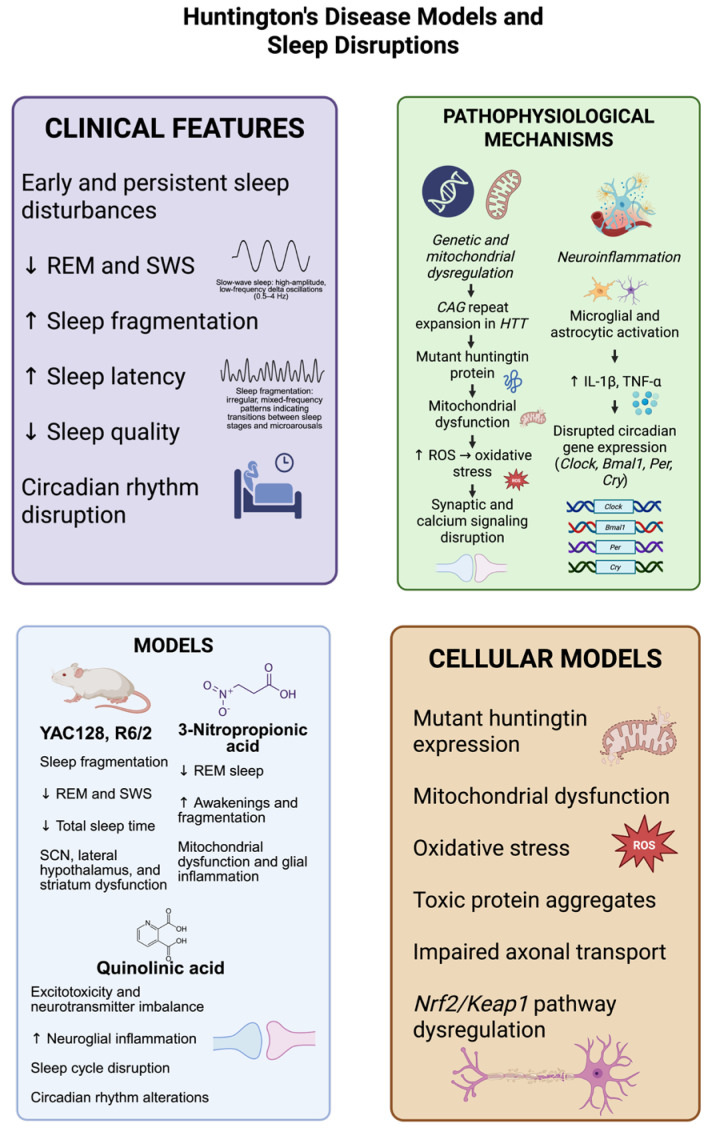
Models of Huntington’s disease (HD) reliably reproduce the major sleep abnormalities seen in the disease, which consist of decreased rapid eye movement (REM) sleep, the loss of slow-wave sleep (SWS), increased sleep fragmentation, prolonged sleep onset latency, and disrupted circadian rhythms. Transgenic mouse models, including YAC128 and R6/2, and neurotoxin-based models, using 3-nitropropionic acid and quinolinic acid, display mitochondrial dysfunction, neuroinflammatory responses, and changes in the brain areas crucial for sleep regulation, such as the suprachiasmatic nucleus, hypothalamus, and striatum. Molecularly, the expression of mutant huntingtin protein driven by CAG repeat expansion results in oxidative stress via an increase in reactive oxygen species (ROS), disturbed synaptic function, and the defective expression of circadian genes (*Clock*, *Bmal1*, *Per*, *Cry*). Complementary cellular models further support these observations, showing the presence of cytotoxic protein aggregates, abnormalities in axonal transport, and the Nrf2/Keap1 antioxidant pathway disruption. These pathophysiological mechanisms together lead to sleep–wake disturbances, (decreased SWS, REM, sleep quality and total sleep time, increased awakenings, fragmentation and sleep latency), which can accelerate neurodegeneration. The dates show ↓ = Decreased; ↑ = Increased. (Created with BioRender.com). By Emiliano Gonzalez and Alexis Ponce, 2025.

**Figure 4 biomedicines-13-02005-f004:**
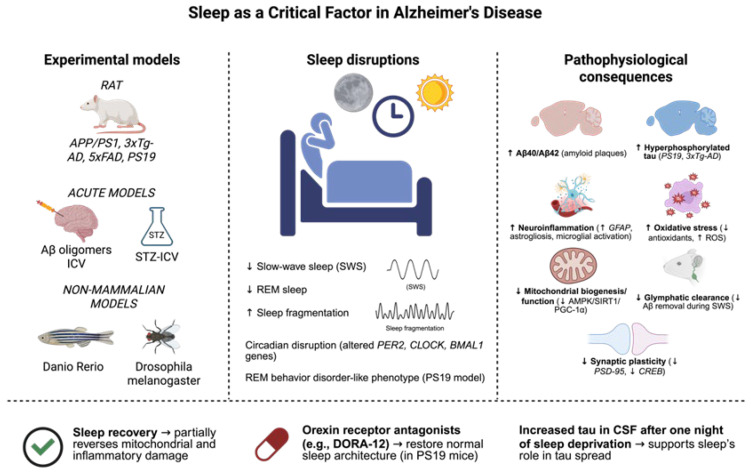
Sleep disturbances are involved in the pathophysiology of Alzheimer’s disease (AD), as they are active participants in the development of the disease and a clinical manifestation of the disease. Several experimental models, including transgenic mouse lines (APP/PS1, 3xTg-AD, 5xFAD, PS19), acute models of Aβ oligomers, or the intracerebroventricular administration of streptozotocin (STZ-ICV) injected into the brain, and non-mammalian models including *Danio rerio* and *Drosophila melanogaster*, have been useful for the reproduction of sleep-related alterations that frequently appear in patients. These consist of decreased slow-wave sleep (SWS) and rapid eye movement (REM) sleep, increased sleep fragmentation, and circadian rhythm disturbances. Equally, some models (e.g., PS19) also demonstrate hallmarks of REM behavior disorder. Notably, sleep deprivation was demonstrated to aggravate several hallmarks of the pathological process, such as Aβ40/42 deposition, tau hyperphosphorylation, neuroinflammation, oxidative stress, mitochondrial dysfunction, glymphatic clearance deficit, as well as synaptic plasticity loss. On the other hand, interventions, e.g., involving sleep recovery or the administration of orexin receptor antagonists (e.g., DORA-12), have been shown to be able to reverse sleep architecture to physiological conditions and reduce cellular damage. These results suggest the possible protective action of sleep and point to its potential as a therapeutic target in Alzheimer’s disease. The dates show ↓ = Decreased; ↑ = Increased (Created with BioRender.com). By Emiliano Gonzalez and Alexis Ponce, 2025.

**Table 1 biomedicines-13-02005-t001:** Summary of sleep disturbances observed in animal models of neurological diseases. Arrows indicate the direction of change in specific sleep parameters based on experimental findings. ↓ = Decreased; ↑ = Increased; ± = Altered; — = Not reported.

*Condition*	*SWS*	*REM*	*Fragmentation*	*Sleep Latency*	*Sleep Duration*	*Sleep Efficiency*	*Wakefulness*	*Circadian Rhythm*
Parkinson’s models	↓	±	↑	—	—	—	—	—
Epilepsy models	↓	↓	↑	↑	↓	↓	↑	±
Huntington’s models	↓	↓	↑	↑	—	—	—	±
Alzheimer’s models	↓	↓	↑	—	—	—	—	—

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
