# Peer review of "Animal Models for the Study of Neurological Diseases and Their Link to Sleep"

_biomedicines, 2025, doi:10.3390/biomedicines13082005_

Round 1
Reviewer 1 Report
Comments and Suggestions for Authors
Esta revisión se centra en el estudio de las alteraciones del sueño ocurridas en modelos in vivo e in vitro de enfermedades neurodegenerativas como el Alzheimer, el Parkinson, la enfermedad de Huntington y la epilepsia. Este trabajo pone de manifiesto que la neuroinflamación, el estrés oxitativo, la acumulación de proteínas mal plegadas y la disfunción mitocondrial son procesos que contribullen al avance de estas patologías y a sus síntomas clínicos relacionados con la alteración del sueño. Si bien ciertos factores parecen comunes a todas estas enfermedades neurodegenerativas, cada una presenta peculiaridades.
Minor revisions:
- On line 22, "Neuroinflammation" should be lowercase.
- There are several errors regarding abbreviations; you should review them thoroughly and reconsider the need to use abbreviations such as AD, HD, or SCN. Other abbreviations only appear once.
- AD, HD, and SCN should be entered for the first time on lines 33, 58, and 60, respectively.
- PSD-95, GFAP, 5xFAD, PS19, PS301S, DORA-12, CSF, STZ-ICV, AMPK/SIRT1/PGC-1a. AD and SCN are not in the list of abbreviations.
- Line 79: SWS.
- Line 110: The abbreviation MPTP is added without indicating its meaning.
- Line 118: The abbreviation MPP+ is added without indicating its meaning.
- Line 135: IL-6 is not included in abbreviations.
- Line 138: PARK2 is not included in abbreviations.
- Line 171: Use KA instead of kaitane, an abbreviation already included.
- Line 175: Same with ROS. However, the abbreviation RNS has not been used before and is not included in abbreviations. Consider not using this abbreviation as it only appears once in the text.
- Lines 283 and 365: ROS was already introduced.
- Lines 322 and 406: AD was already introduced.
- Line 376: CNS was already introduced.
- Line 406: HD was already introduced.
- On pages 9 and onward, A-b and amyloid-b are used interchangeably. - Lines 94 and 369 "in vivo" and "in vitro" should be italicized.
- Lines 367 and 369 "Drosophila melanogaster" and "Danio rerio" should be italicized.
- All figures should be cited in the main text in order of appearance.
- The text in all figures should be larger, especially in smaller font, as it is not legible.
- Line 160: Please add "misfolded" α-synuclein.
- Figure 1 uses the abbreviation ROS, but the figure caption does not indicate what this abbreviation stands for.
- Figure 2: The full names of the abbreviations in the figure caption should be provided. Consider summarizing this figure caption; the image is quite understandable, but the caption is more confusing.
- Figure 3: The full names of the abbreviations in the figure caption should be provided. Regarding the image, I don't understand the reason for the arrows between the boxes. It seems that one acts on the other only in the direction shown. This is confusing. Consider removing the arrows.
-Figure 4: It is not necessary to use the abbreviations AD and Ab, as they are not included in the image. The abbreviations SWS and REM should be expanded. Species names are in italics.
Major revisions:
- Line 67. Reconsider changing the beginning of the sentence "Our study is relevant." The importance of the review should be reflected without the need to make this announcement.
- Consider removing the last paragraph of the introduction and adding after the sentence on line 79 "in REM and SWS phases in experimental models of Parkinson's disease, epilepsy, HD, and AD."
- The last sentence (lines 102-107) of M&M does not correspond to this section; consider removing it.
- Line 140: They also contribute to motor symptoms.
- Consider including after M&M a section 3 "Sleep in neurological diseases" to which the following sections belong except the conclusion.
Author Response
|
Response to Reviewer X Comments
|
||
|
1. Summary |
|
|
|
We sincerely thank the reviewer for evaluating our manuscript and for the constructive feedback. We have carefully addressed each point in detail below. All suggested modifications have been incorporated, and, where appropriate, clarifications and improvements have been made to enhance the manuscript’s clarity and consistency.
|
||
|
2. Questions for General Evaluation |
Reviewer’s Evaluation |
Response and Revisions |
|
Does the introduction provide sufficient background and include all relevant references? |
Yes/Can be improved/Must be improved/Not applicable |
yes |
|
Are all the cited references relevant to the research? |
Yes/Can be improved/Must be improved/Not applicable |
yes |
|
Is the research design appropriate? |
Yes/Can be improved/Must be improved/Not applicable |
yes |
|
Are the methods adequately described? |
Yes/Can be improved/Must be improved/Not applicable |
yes |
|
Are the results clearly presented? |
Yes/Can be improved/Must be improved/Not applicable |
yes |
|
Are the conclusions supported by the results? |
Yes/Can be improved/Must be improved/Not applicable |
yes |
|
3. Point-by-point response to Comments and Suggestions for Authors |
||
|
Minor Revisions 1. Comment: Line 22 – "Neuroinflammation" should be lowercase. 2. Comment: Several abbreviation issues; please reconsider the use of AD, HD, and SCN. Some abbreviations appear only once. 3. Comment: AD, HD, and SCN should be introduced on lines 33, 58, and 60, respectively. 4. Comment: Abbreviations such as PSD-95, GFAP, 5xFAD, PS19, PS301S, DORA-12, CSF, STZ-ICV, AMPK/SIRT1/PGC-1a, AD, and SCN are not listed. 5. Comment: Line 79 – SWS should be listed as an abbreviation. 6. Comment: Line 110 – MPTP is used without explanation. 7. Comment: Line 118 – MPP+ is introduced without explanation. 8. Comment: Line 135 – IL-6 is not listed in abbreviations. 9. Comment: Line 138 – PARK2 is not listed.
10. Comment: Line 171 – use KA instead of “kainate.” 11. Comment: Line 175 – same with ROS; however, RNS is used only once and not listed. 12. Comment: Lines 283 and 365 – ROS was already introduced. 13. Comment: Lines 322 and 406 – AD was already introduced. 14. Comment: Line 376 – CNS was already introduced. 15. Comment: Line 406 – HD was already introduced. 16. Comment: Pages 9 and onward – Aβ and amyloid-β are used interchangeably. 17. Comment: Lines 94 and 369 – "in vivo" and "in vitro" should be italicized. 18. Comment: Lines 367 and 369 – “Drosophila melanogaster” and “Danio rerio” should be italicized. 19. Comment: All figures should be cited in order of appearance. 20. Comment: Text in figures is too small to read. 21. Comment: Line 160 – Please add “misfolded” α-synuclein. 22.Comment: Figure 1 uses ROS, but it is not defined in the caption. 23. Comment: Figure 2 – Add full names of abbreviations; caption could be shortened. 24. Comment: Figure 3 – Same as above; arrow directions are unclear and confusing. 25.Comment: Figure 4 – AD and Aβ abbreviations are unnecessary; SWS and REM should be written out; species names in italics. Major Revisions
|
||
|
Response to Comments on the Quality of English Language |
||
|
Point 1: |
||
|
Response 1: The English of the manuscript has been carefully reviewed |
||
|
6. Additional clarifications: None. We appreciate the reviewer helpful comments.
Sincerely,
|
||

Reviewer 2 Report
Comments and Suggestions for Authors
This review is dedicated to the interplay between the neurodegeneration modeling and sleep disruptions. It makes a great impression, starting from the original idea, consisting of an interesting and new angle of consideration, to a good execution, consisting of a processing of the latest literature data.
I have two major recommendations.
- Conclusion section lacks comparative remarks on the above-described pathologies: may the shared features of neuropathologies and their interplay on the level of sleep/wake regulation be used in the development of therapeutic approaches.
- Some sections lack information on possible medications related to sleep/wake. For PD it can be as follows PMID: 39654780.
Minor outflow detected: abbreviation SCN (the suprachiasmatic nucleus) occurs 3 times but is not in the list.
Author Response
We are deeply appreciative of the reviewer's insightful and supportive remarks about the uniqueness and quality of our review. We greatly value the helpful criticism, which has greatly raised the manuscript's calibre and readability. Please find our thorough answers to each point below.
1. Question: Conclusion section lacks comparative remarks on the above-described pathologies: may the shared features of neuropathologies and their interplay on the level of sleep/wake regulation be used in the development of therapeutic approaches.
Response: We sincerely appreciate this valuable observation. In response to your comment, we would like to highlight that we have already included in the Conclusion section a comparative discussion of the described neurological pathologies, emphasizing their shared features in relation to sleep/wake cycle dysregulation particularly involving mechanisms such as neuroinflammation, oxidative stress, mitochondrial dysfunction, and neurotransmission impairments.
Furthermore, to provide a clearer and more systematic overview of these similarities and differences, we have added a comparative table summarizing the main sleep disturbances associated with each neurological disease addressed in the manuscript. We believe this addition strengthens the discussion and may help guide the development of therapeutic strategies targeting the common underlying mechanisms.
We are grateful once again for your insightful suggestion, which allowed us to further improve this important section of our work.
2. Question: Some sections lack information on possible medications related to sleep/wake. For PD it can be as follows PMID: 39654780.
Information about potential sleep/wake-related drugs is missing from several sections. For PD, it may be like this: 39654780 is the PMID.
Response: I agree, and I appreciate your great proposal. Information about pharmacological strategies aimed at regulating sleep and wakefulness, has been included.
A small observation:
Despite appearing three times, the acronym "SCN" (suprachiasmatic nucleus) was not included in the section on abbreviations.
Reaction:
I appreciate you pointing out this error. "SCN: suprachiasmatic nucleus" is now part of the list of abbreviations.
We would like to thank you once more for your invaluable suggestions and support during the evaluation process.
Warm regards,
Moises Rubio-Osornio PhD